# Breeding Buckwheat for Nutritional Quality in the Czech Republic

**DOI:** 10.3390/plants10071262

**Published:** 2021-06-22

**Authors:** Dagmar Janovská, Michal Jágr, Pavel Svoboda, Václav Dvořáček, Vladimir Meglič, Petra Hlásná Čepková

**Affiliations:** 1Gene Bank, Crop Research Institute, Drnovská 507/73, 161 06 Prague 6, Czech Republic; janovska@vurv.cz; 2Quality and Plant Products, Crop Research Institute, Drnovská 507/73, 161 06 Prague 6, Czech Republic; jagr@vurv.cz (M.J.); dvoracek@vurv.cz (V.D.); 3Molecular Genetics, Crop Research Institute, Drnovská 507/73, 161 06 Prague 6, Czech Republic; pavel.svoboda@vurv.cz; 4Crop Science Department, Agricultural Institute of Slovenia, Hacquetova ulica 17, SI-1000 Ljubljana, Slovenia; vladimir.meglic@kis.si

**Keywords:** breeding, common buckwheat, *Fagopyrum*, morpho-agronomic traits, protein, phenolic compounds, mass spectrometry

## Abstract

Buckwheat is a nutritionally valuable crop, an alternative to common cereals also usable in gluten-free diets. The selection of buckwheat genotypes suitable for further breeding requires the characterization and evaluation of genetic resources. The main objective of this work was to evaluate selected phenotypic and morphological traits using international buckwheat descriptors, including total phenolic content and antioxidant activity, on a unique set of 136 common buckwheat accessions grown in 2019–2020 under Czech Republic conditions. In addition, UHPLC-ESI- MS/MS was used to analyze a wide spectrum of 20 phenolic compounds in buckwheat seeds, including four flavanols, three phenolic acids, seven flavonols, four flavones, and two flavanones. Significant differences among years and genotypes were observed for morphological traits (plant height and 1000-seed weight) and antioxidant activity, as well as levels of observed chemical compounds. Antioxidant activity, crude protein content, plant height and rutin content were characterized by higher mean values in 2020 than in 2019 and vice versa for total polyphenol content and 1000-seed weight. Crude protein content was the most stable across years, while total polyphenol content and rutin content varied greatly from year to year. The most abundant phenolic compounds were rutin, hyperoside, epicatechin, catechin, vitexin, isovitexin, orientin and isoorientin. Protein content was negatively correlated with plant height, catechin and epicatechin content. On the other hand, AA and TPC were positively correlated with rutin, hyperoside and chlorogenic acid. Five accessions showed high stability of the evaluated traits under changing conditions within both years of observation. These materials can be used in breeding programmes aimed at improving buckwheat genotypes with emphasis on quality traits.

## 1. Introduction

Neglected or underutilized crops, such as minor millets and pseudo-cereals, were part of the common diet of ancient cultures. However, since the Green Revolution, rice, wheat and maize have provided more than 60% of caloric intake and they have replaced traditional crops, resulting in a nutrient-poor diet [1]. Among pseudo-cereals, there is buckwheat (*Fagopyrum* Mill. Polygonaceae), a small genus including less than 30 species, mainly endemic to southern China [2], of which only two, common buckwheat and Tartary buckwheat, are used for food purposes [3]. Common buckwheat (*Fagopyrum esculentum* Moench) is a dicotyledon, annual crop originated from Yunnan province in China [4]. It is an important crop in mountain regions in Himalayan countries, China, Korea, Japan, Russia, USA, Ukraine and other parts of Europe [5]. Buckwheat has been traditionally used as food for humans [6]. Buckwheat grains contain a variety of nutrients [7,8]. Protein content in common buckwheat in seed varies from 8.51% to 18.87%, depending on variety [9] and with low content of prolamin [10]. Buckwheat flour contains 70–91% (*w*/*w*) of starch depending on milling method [11]. The content of fat is close to 3% whereas the fibre concentration is high (~15%) [12]. In particular, buckwheat is rich in K, Mg, and Ca [13], Fe and Zn concentrations are considered higher than in main crops [14]. The main phenolic compounds in four buckwheat varieties were confirmed to be rutin, vitexin, isovitexin and hyperoside [15]. Moreover, in several research publications, buckwheat germplasms confirmed great diversity in morphological and agronomic traits [16,17,18], likewise in nutritional [19] as well as health-promoting contents [15,20,21]. Further, many scientific publications/reviews have recently published summary information on various positive effects of buckwheat on human health [22,23,24,25].

Buckwheat has been bred in many countries as a traditional crop, resulting in many new varieties since the 1920s [8]. In the last 30 years, research on buckwheat genetic resources has been proceeded with great attention to valuable traits important for quality improvement. Currently, more than 10,000 samples of buckwheat genetic resources are conserved and stored in various gene banks worldwide [3].

Genetic resources stored and conserved in gene banks should be evaluated and properly characterized to find out what useful and valuable genetic diversity traits they contain. Characterization and evaluation of genetic resources should be done before selecting the right material for a particular purpose, for example breeding [26]. It is necessary to use an effective methodology for describing accessions in the same collection to distinguish them [27]. To date, there are only a limited number of comprehensive studies on buckwheat genetic resources and their adaptation to different environments. One of the most extensive studies described 857 germplasm accessions, including more than six *Fagopyrum* species cultivated and evaluated in India [28]. Suitability of buckwheat cultivation in the Mediterranean environment was tested in four buckwheat varieties in southern Italy [29]. Recently, the results of a study conducted in the UK were published, evaluating two varieties of common buckwheat under the cool temperate climate of North-East England [14].

The main objective of this study was to evaluate a unique collection of 136 common buckwheat accessions from gene banks according to agro-morphological traits and content of selected nutrients in 2019–2020 under the conditions of the Czech Republic. The best identified accessions and information about them will help buckwheat breeders to select the most suitable material for conditions in Central Europe.

## 2. Results and Discussion

### 2.1. Weather Conditions

In general, 2019 can be considered warmer and drier compared to 2020, which was characterized by abundant rainfall and monthly average temperatures slightly above the 30-year average (except May). 2019 had much warmer summer months (June and July). Average precipitation in 2020 was quite variable; in most cases rainfall was close to the 30-year average, but in March and October 2020, for example, average precipitation was almost double the 30-year averages. On the other hand, April and July 2020 had significantly less average precipitation. Rainfall in 2019 was similar to the 30-year average, with the exception of September, where precipitation was one third higher and in July where precipitation was half. Buckwheat prefers cooler rather than warmer summers, and first sowing times generally produced a higher content of most polyphenols [29].

### 2.2. Morphological Evaluation

The value of plant genetic resources depends on evaluation data to promote their use. In this study, we used selected descriptors from the international list of descriptors for buckwheat published by IPGRI [30].

136 accessions of common buckwheat were evaluated under field conditions in 2019 –2020. The results obtained in both years are summarized in the text below and in Appendix A. Assessment of genetic variability within and between plant populations is routinely performed using various techniques. One of these is morphological evaluation using visually accessible traits, such as growth habit and seed shape [31]. The morphological traits presented in this study are 1000-seed weight (TSW) and plant height (PH). TSW results showed a significant difference between 2019 and 2020, with mean TSW value in 2019 (24.82 ± 3.63 g) being 8 g higher than in 2020 (16.29 ± 4.38 cm) (Figure 1). In the Indian buckwheat collection where mean value of TSW was 20.56 g, which is in the middle of mean values obtained in our study for two years in the locality in the Czech Republic for two years [28]. The maximum mean value in 2019 was reached in accession ’Marta´ (35.58 ± 0.05 g) and the minimum value was in accession ´Sperli´ (15.03 ± 0.67 g). Mean values were also much lower in 2020; the maximum mean value (29.21 ± 0.00 g) was in ´Jana´ and the minimum mean value (5.48 ± 0.00 g) was in ´Temnica na Krasu´. The maximum difference (19.98 g) in TSW compared to the previous year was found in ´Billy´. Only two accessions responded with a slight increase in TSW in 2020 compared with 2019´Jana´ and ´MC 039´. The mean values of TSW obtained in our study were consistent with previously published TSW values [16,32,33]. In 2019, 86% of accessions had TSW in the range of 20–30 g, which was consistent with the findings of Rauf et al. [16] and Zhou et al. [3]. However, 2020 was significantly less favourable for grain formation and only 20% of the accessions reached TSW in the range of 20–30 g, with most of the accessions having TSW below 20 g. The results of TSW in 2020 were probably influenced by adverse weather conditions when plant growth was affected by low temperatures during the post-sowing period (May to July). Similarly, Domingos and Bilsborrow [14] measured TSW in two buckwheat varieties at the level of 21.1 vs. 21.1 g in the cool temperate climate of the UK. Ghiselli et al. [17] pointed out the strong influence of locality and buckwheat variety on TSW. Characterization of 136 buckwheat accessions revealed high diversity in TSW between years and accessions, which was also statistically confirmed by Rauf et al. [16]. Such diversity confirmed in TSW may be useful for selecting suitable material for diverse climatic conditions; for instance, Domingos and Bilsborrow [14] observed that buckwheat varieties with uniform maturation and larger seeds were better suited to cool temperate climates.

Plant height (PH) was another morphological trait evaluated in the buckwheat collection grown in field trials. Buckwheat accessions had a wide range of PH and significant differences were observed for year and for accessions (Figure 1). Among the total number of 136 evaluated accessions, 91 accessions reached higher PH values in 2020. Mean PH value (112.11 ± 18.52 cm) for 2020 was slightly higher than for 2019 (106.64 ± 11.00 cm), while it was consistent with the mean PH value, which was 107.40 cm for 964 evaluated common buckwheat genotypes in China [3] but less than observed in buckwheat genetic resources in India [28]. The maximum mean PH was reached by ´La Harpe´ (154.60 ± 27.40 cm) in 2019 and by ´Česká krajová´ (157.30 ± 12.95 cm) in 2020. The minimum mean values were seen in accession ´MC 060´, at 78.00 ± 4.94 cm in 2019, and ´Krupinka´ in 2020, with an even lower height of 68.50 ± 3.30 cm. Moreover, one of the common breeding goals related to PH is lodging resistance and breeding new semi-dwarf common buckwheat varieties [34]. PH can also be regulated by plant density and level of nitrogen fertilization, which improve photosynthetic capacity and increase production [35]. PH was significantly positively correlated with seed yield [36] and with seed weight [28] but in our case we observed slightly negative correlation (−0.20) with TSW.

### 2.3. Crude Protein Content

Further evaluation of the buckwheat collection involved selected constituents of nutritional value and traits associated with the study objective of selecting excellent plant material for future breeding. Crude protein (CP) content is one of the key components of buckwheat from a nutritional point of view. Mean CP content was very similar in both years (14.82 ± 1.46% and 13.84 ± 0.62% dry weight (d.w.), respectively). The results obtained are in agreement with protein content detected in buckwheat cultivated in southern Italy, where the highest value (14.1%) was obtained in plots under irrigation [29]. It is a well-known fact that protein content depends on the variety and environmental conditions; here the protein content was stable in both years and most accessions had a slightly higher CP content in 2020 than in 2019, with a mean annual difference of only 0.99% (Figure 2). This finding was observed in an experiment done in the UK [14]. Above that, statistically significant differences in CP content between varieties and years were confirmed. Similarly, buckwheat cultivated under organic cultivation in mountainous regions of Italy where variation in protein content within years was observed [17]. The authors of many previous studies referred to more or less similar mean values of protein content [37,38]. The obtained data are in agreement with a previous one-year study conducted at the same location in Prague Ruzyně [39] and very close to the results of cultivated common buckwheat recently grown in Poland [10], as well as slightly higher than results of an experiment done in the UK [14]. The maximum mean value of CP content was obtained by ´MC044´with 17.93 ± 0.44% d.w. in 2019 and ´MC039´, with 21.10 ± 0.30% d.w. The near maximum value was reached during an experiment in Slovakia [9]. The lowest CP content was recorded in ´Panda´ (12.46 ± 0.03% d.w.) in 2019 and in ´Chishiminskaya´ (13.26 ± 0.29% d.w.) in 2020. The mean values of CP content observed in previous studies were at lower levels of 11–12% d.w., the same as in common wheat [12,40]. On the other side, according to several authors [12,41,42], buckwheat grains contain high-quality protein rich in lysine and a balanced content of other amino acids with a biological value of up to 93%. However, the digestibility of buckwheat proteins in the small and large intestine is reduced by the polyphenols naturally present in the grains [25,43]. In this study, protein content was negatively correlated with catechin (−0.39) and epicatechin (−0.39), and weakly negatively correlated with PH (−0.17).

### 2.4. Total Phenolic Content and Antioxidant Activity

Total phenolic content (TPC), antioxidant activity (AA) and 20 polyphenols content were evaluated in the buckwheat collection (Appendix A). In 2020, higher mean values of TPC prevailed across accessions but in 57 accessions, the trend was the opposite, showing higher mean values of TPC in 2019 (Figure 3). The mean value of TPC in 2020 was higher by 41.79 GAE g/kg d.w. compared to 2019. The maximum difference between the years (444.06 GAE g/kg d.w.) was recorded in the accession ’Sweden-1’; this accession also attained the highest TPC value in 2020 (920.00 ± 10.52 GAE g/kg d.w.).

The maximum value in 2019 was recorded in the accession ’Aiva’, at 552.43 ± 5.11 GAE g/kg d.w. The minimum TPC value, and also the lowest value in the whole collection for both years, was found in ’Feniks’ in 2020 at 72.85 ± 2.62 GAE g/kg d.w. These results supported the opinion of Martinéz-Villaluenga et al. [23] that buckwheat is one of the best sources of phenolic compounds. Here, the correlations between TPC and flavonoid compound were calculated using Spearman’s correlation coefficient. TPC had a positive correlation at around 0.30 with AA, rutin and quercetin, isoquercetin, hyperoside, orientin, isorientin, procyanidine B2 and chlorogenic acid (Figure 4).

AA of all 136 buckwheat accessions determined with the stable radical 2,2-diphenyl-1-picrylhydrazyl (DPPH) is shown in Appendix A. On average, AA was higher in 2020 than in 2019; only 27 accessions showed the opposite trend in 2019 (Figure 3). As mentioned above, lower temperatures were recorded in Prague Ruzyně in 2020 than in the previous year which could lead to higher antioxidant concentration due to significantly higher plant population at harvest, resulting in increased interception of radiation [14]. Among all the accessions, the highest value in 2020 was determined ‘Sadkom’ (33.12 ± 0.53 mmol TE/g d.w.) and the lowest AA in ‘Eva’ (10.45 mmol TE/g d.w.). AA in 2019 ranged from 7.52 ± 0.10 mmol TE/g d.w. (‘Grechka’) to 25.77 ± 2.44 mmol TE/g d.w. (‘La Harpe’). In buckwheat, rutin is considered a major contributor to AA [44]. Spearman’s correlation results supported the theory of positive correlations between AA and all tested flavonoids. In particular, the highest values were estimated between AA and rutin (0.42), hyperoside (0.40), quercitrin (0.46) and chlorogenic acid (0.47) (Figure 4). Similarly, a significant positive correlation for rutin with TPC was confirmed [45]. Previous studies have shown the dependence of AA on the specific AA of each phenolic compound as well as TPC [46].

### 2.5. Phenolic Compounds Content

The sum of phenolic compounds (PHE) was evaluated in the buckwheat collection. The difference between years and between accessions was also confirmed for this trait as in Ghiselli et al. [17]. The mean value for 2020 was higher than 2019 by 166.74 ug/g d.w. There was an almost uniform trend across all 136 buckwheat accessions; higher values were recorded in 2020, with only 20 samples reaching the higher sum of phenolic compounds in 2019. The highest value was recorded in ’Tempest’ (1362.30 ± 19.61 ug/g d.w.) in 2020 and in ‘Emka II’ in 2019 (1045.86 ± 4.75 ug/g d.w.). ‘HUNORS 2010-9’ had the least sum of phenolics (233 ± 6.10 ug/g) in 2019 and ‘Panda I’ in 2020 (313.10 ± 4.63 ug/g d.w.). A significant increase in the sum of phenolic compounds from year to year was observed in ‘CD8217’ (849.52 ug/g d.w.), ‘Tempest’ (830.43 ug/g d.w.), ‘PA056’ (609.22 ug/g d.w.) and ‘Nostran’ (531.09 ug/g d.w.). As described in Materials and Methods, the weather conditions in 2020 were different from those in 2019, which could be related to the higher flavonoid production in 2020 as a plant response to the adverse weather conditions. It is known that flavonoids are especially important for plant resistance to changing environmental conditions and for plant growth and development [47]. Siracusa et al. [29] found that water stress may also result in a general increase in polyphenols levels.

Although various methods have been used in the past to detect phenolic compounds in buckwheat grains, flour and various plant parts [15,21,45,48,49], use of the very sensitive and susceptible PRM method of UHPLC-HRMS/MS has not been published yet. A total of 20 phenolic compounds were determined in our study, of which four were flavanols, three were phenolic acids, six were flavonols, five were flavones, and two were flavanones. The identified phenolic compounds were quantified by analysis; detailed information about all the identified compounds in buckwheat accessions are provided in Appendix A. The most abundant flavonoids in buckwheat seeds were rutin, epichatechin, catechin, vitexin, isovitexin, orientin and isoorientin. Lower contents were determined for gallic acid, chlorogenic acid, procyanidin B2, isoquercetin, quercetin and quercitrin. Caffeic acid, procyanidin B1 + B3, apigenin, kaempferol, naringenin and hesperidin were present at trace levels in buckwheat grains in both years.

The members of the *Fagopyrum* genus are promising sources of flavonoids. Isoquercetin, together with rutin, is a known major phenolic compound in common buckwheat and Tartary buckwheat, with a wide range of contents depending on genotype and collection area [16] and correlates with cumulative solar radiation [14].

Of the flavonoids studied, the highest amount was confirmed for the flavonol rutin in buckwheat seeds across all samples studied. Rutin or vitamin P is the quercetin-3-O-rutinoside and is valued for its antioxidant, angioprotective, antibacterial and hepatoprotective properties [47] with potential therapeutic applications in the treatment of Alzheimer’s disease [50] and antiviral activity along with emodin [43]. Almost all the accessions in the collection were characterized by higher rutin content in 2020, with the exception of 11 accessions (Figure 3). The mean value of rutin content in 2020 (224.16 ± 70.66 ug/g d.w.) was higher than in 2019 (186.92 ± 64.18 ug/g d.w.), which is less than the values observed by Kalinova and Vrchotova [51] and significantly higher than that (0.064 mg/g) obtained during summer cropping in the UK [14]. On the other hand, rutin content in buckwheat grains cultivated in Slovakia was much higher [52] and closer to the values more common for Tartary buckwheat seed Borovaya and Klykov [53] and sprouted seeds [54]. Surprisingly, the highest rutin content across all accessions was reached by the ‘Nostrano’ accession (819.89 ± 16.66 ug/g d.w.) in 2019, which also showed the highest decrease (by 545.04 ug/g d.w.) in rutin content from 2019 to 2020. The maximum content of rutin in the second year was in the accession ‘CD 8217’, at 627.35 ± 3.90 ug/g d.w. The minimum value of rutin was 64.53 ± 1.22 ug/g d.w. (‘Vrhtrebnje’) in 2019 and 104.82 ± 1.85 ug/g d.w. (‘Panda I’) in 2020. ‘Nostrano’ probably strongly reflected environmental conditions in 2019 by synthesizing secondary metabolites for its defence. However, the rest of the accessions responded in the opposite way, producing more secondary metabolites in response to stress conditions during growth in 2020, which could indicate an individual genotype ability to adapt to changes in the environment. When comparing the specific values obtained in ‘Čebelica’, ‘Bamby’, ‘La Harpe’ and ‘Špačinská’ with those published by Kiprovski et al. [21], we obtained significantly higher values of rutin. This could be influenced by location and growing season.

The flavonol quercetin is of particular interest when it comes to buckwheat grains and products [25]. In the present study, this flavonol was detected together with isoquercetin and quercitrin in buckwheat samples, but in much lower amounts than rutin. Here, evaluation of 136 buckwheat accessions revealed the amount of these compounds in the order isoquercetin > quercetin > quercitrin. The mean content of isoquercetin was higher in 2020 (9.79 ± 5.51 ug/g d.w.) than in 2019 (6.43 ± 4.24 ug/g d.w.). The highest value for isoquercetin was in ‘Rubra’ (34.12 ± 1.93 ug/g d.w.) in 2019 and ‘UC0100282’ (37.14 ± 2.89 ug/g d.w.) in 2020. The mean values for quercetin were 3.25 ± 5.81 ug/g d.w. and 7.66 ± 4.31 ug/g d.w. in 2019 and 2020, respectively. An unusually high amount of quercetin (66.88 ± 0.58 ug/g d.w.) was detected in ‘Temnica na Krasu’ in 2020. However, this was an exception among all the studied buckwheat samples and the other values observed were more or less at the same level but at a lower level in comparison with data presented by Vollemannová et al. [52]. Quercitrin in the buckwheat collection was found at a level of 3.87 ± 2.66 ug/g d.w. for 2020 and 1.19 ± 0.98 ug/g d.w. for 2019. The highest value for 2020 was shown by ‘Temnica na Krasu’, with 17.93 ± 0.51 ug/g d.w.

Vitexin was the most abundant compound in the flavone group and the second most abundant phenols of all those detected in this study. Previously, vitexin was also confirmed as one of the most abundant compounds in buckwheat grains [21,29]. Like rutin content, vitexin content was observed to be higher in the majority of buckwheat samples in 2020; only 38 accessions showed opposite trend. The content of vitexin ranged from 3.39 ug/g d.w. to 130.57 ug/g d.w. for 2019 and from 18.22 ug/g d.w. to 174.57 ug/g d.w. for 2020. The highest increase compared to the previous year was observed in ‘Mestnyy 31’ (101.63 ug/g d.w.); on the other hand, the highest decrease was observed in ‘Pyra’ (32.22 ug/g d.w.). Slightly lower values were recorded in the isomer of vitexin: isovitexin in buckwheat samples. The mean values between years differed and were similar to vitexin, accessions showed higher values of isovitexin in 2020. The obtained results are comparable to the values published by Vollmannova et al. [52], who reported 0.010–0.212 mg/g d.w. The content of vitexin in buckwheat grains cultivated in Slovakia, reported the lowest value in the variety ‘Ballada’ (0.010 mg/g d.w.) but this is not in line with our findings where ‘Ballada’ had stable and almost six times higher content of vitexin, higher in both experimental years (66.91 ± 1.34 ug/g d.w. for 2019 and 53.50 ug/g d.w. for 2020). However, in the case of variety ‘Pyra’, the results observed in both years were in agreement with those reported by Vollmannova et al. [52]. Furthermore, a significantly wider range of vitexin content was measured in the 136 tested accessions compared to Kiprovski et al. [21] with a range of vitexin content between 0.003–0.033 mg/g d.w. The mean value of isovitexin for 2019 was 27.92 ± 12.04 ug/g d.w. and for 2020 it was 47.93 ± 19.87 ug/g d.w. In 2019, the range of values obtained was between 2.19 ug/g and 74.41 ug/g and in 2020 it was between 13.85 ug/g d.w. and 125.37 ug/g d.w. The highest jump in the mean value of isovitexin from year to year was in ‘Tempest’, the value increased by 84.87 ug/g d.w. in 2020. The other two flavones, orientin and isoorientin (or homoorientin), were also among the abundant flavonoids in buckwheat grains. Here, the trend of higher contents in 2020 was observed in all buckwheat samples studied. The highest content of orientin was possessed by ‘Emka II’ (60.47 ± 1.27 ug/g) in 2019 and ‘Tempest’ (110.70 ± 1.35 ug/g d.w.) in 2020. ‘Sadkom’ made the greatest jump with an increase in orientin from 13.67 ± 0.15 ug/g d.w. to 70.53 ± 0.96 ug/g d.w. from year to year. An analogous situation was observed in the content of isoorientin. The highest content of isoorientin was found in ‘Emka II’ (56.19 ± 1.03 ug/g d.w.) in 2019 and ‘Tempest’ (76.54 ± 1.42 ug/g d.w.) in 2020 and the highest difference between years was observed in the content of isoorientin in ‘Sadkom’, at 55.59 ug/g d.w. According to present knowledge, isovitexin exerts similar pharmacological effects as vitexin, partly due to their similar chemical structure. Isovitexin and vitexin could be suitable potential therapeutic candidates for many diseases or syndromes [55].

The third most abundant flavonoid in buckwheat grains was hyperoside, as in an experiment in southern Italy [29]. The detected levels were also higher in 2020. Hyperoside was detected in samples ranging from 7.44 ug/g to 118.16 ug/g in 2019 and from 15.32 ug/g to 171.12 ug/g in 2020. In the majority of the samples, hyperoside content was found to increase in 2020 as compared to 2019.

The group of the most abundant flavonoids in buckwheat grains included two flavanols, namely catechin and epicatechin. The mean values of epicatechin (52.14 ± 19.42 ug/g for 2020 and 54.46 ± 22.57 ug/g in 2019) were almost double that in catechin (21.75 ± 8.39 ug/g in 2020 and 25.45 ± 10.01 ug/g in 2019) in both years. In catechin, unexpectedly higher values were recorded in 48 accessions for the year 2019. In the case of epicatechin, more than half of the accessions (73) showed a similar trend. The highest value of epicatechin in 2019 was in ‘Iwate Zairai’, at 152.73 ± 3.42 ug/g, while in 2020 it was ‘Chishiminskaya’ at 118.52 ± 1.09 ug/g. ‘Iwate Zairai’ had the greatest decrease in epicatechin (96.50 ug/g) from year to year. The content of catechin appeared to be more stable across buckwheat accessions, without such large fluctuations between years. ‘Krasnostreletskaya’ showed the greatest difference in the content of catechin (50.74 ug/g) between years. ‘Dozhdik’ had the highest content of catechin (69.99 ± 0.72 ug/g) in 2019 and ‘Monori’ (53.66 ± 1.33 ug/g) in 2020. Buckwheat groats with an antilipoperoxidative effect had higher content of epicatechin and catechin [56]. Here, epicatechin and catechin showed negative correlation (−0.39) with protein content. Morishita et al. [57] developed materials with high epicatechin content because of its high contribution to AA.

Other flavanols were also confirmed in buckwheat grains, namely three dimers of procyanidin. The highest content was observed for procyanidin B2, where the levels detected in seeds ranged from 1.90 ± 0.07 ug/g to 12.63 ± 0.13 ug/g in 2019 and from 2.03 ± 0.01 ug/g to 20.59 ± 0.04 ug/g in 2020. The levels of procyanidin B1 + B3 were even lower than those of procyanidin B2. Procyanidin B1 + B3 was proved to be the dominant compound in the seed coat [15], and on the other hand, procyanidin B2 was identified as dominant in the flour of the French variety ‘La Harpe’ [56].

In general, three phenolic acids evaluated in buckwheat grains in our study were found in significantly lower amounts than the polyphenols mentioned above. The most abundant was gallic acid, the mean value of which was 7.67 ± 3.33 ug/g d.w. for 2020 and slightly lower at 6.45 ± 2.82 ug/g d.w. for 2019. The maximum value for 2019 was recorded in ‘MC039’ at 14.18 ± 0.08 ug/g d.w. and for 2020 in ‘Zita’ at 20.72 ± 0.23 ug/g d.w. of gallic acid. However, the values obtained were much lower than those found in Chinese buckwheat genotypes [58]. Here, more than half of the accessions had higher levels of gallic acid in 2020. The content of chlorogenic acid ranged from 0.24 ug/g to 6.79 ug/g d.w. in 2019 and from 0.55 ug/g to 22.35 ug/g d.w. in 2020. Even lower amounts for caffeic acid were estimated across all buckwheat accessions. The mean value for 2019 was 0.25 ± 0.07 ug/g d.w. and for 2020 it was 0.46 ± 0.16 ug/g d.w. In buckwheat samples cultivated in Poland, similar levels were detected for caffeic acid and additionally for ferulic acid, coumaric acid, syringic acid and vanilic acid; but, in general, much higher content was detected in Tartary buckwheat than in common buckwheat [10].

Of the flavanols, the levels of apigenin and kaempferol were also determined, but the levels found were very low. Small amounts of kaempferol were also detected in eight buckwheat varieties cultivated in Slovakia [52], but levels are higher when compared to our data from the Czech Republic. From Slovakia the variety ‘Ballada’, for example, had from 0.01 to 0.03 ug/g d.w. and 1.438 mg/g d.w. [52]. In contrast to our results is the result of HPLC characterization of Chinese buckwheat samples, where apigenin and kaempferol were found at high levels, which could be due to differences in genotypes and environmental factors affecting growth [58]. Kaempferol has been detected in *F. tataricum* and *F. cymosum* and furthermore in low levels in buckwheat microgreens [59]. The effect of sowing time and irrigation on levels of kaempferol was not confirmed [29]. Numerous pharmacological activities have been attributed to apigenin, including anti-inflammatory, antitoxic and anticancer activities [16]. However, the presence of apigenin is more typical of fresh foods, especially vegetables and herbs [60]. Similarly, the low content of the flavanones hesperidin and naringenin was found in buckwheat grown in the Czech Republic. Hesperidin was also detected at low levels in buckwheat herbal tea from leaves [61]. The reason for the low levels could be that hesperidin and naringenin, which both have antioxidant and anti-inflammatory properties, are common glycosides identified mainly in citrus species [62].

The level of all observed descriptors for 136 buckwheat genotypes is shown in Figure 5 in the form of a heatmap based on the percentage maximum values.

As shown in the heatmap, the score values based on the weighted average of the percentage values divide the buckwheat accessions into three distinct groups. These groups include nine accessions with low average scores (≤40%) in both years, 98 accessions with scores between 40 and 50%, and 24 accessions with the highest average score over both years (above 50%), with ‘Tempest’, ‘Emka-II’, and ‘Jana CV-II’ at the top. Of particular interest are nine accessions that scored above 50% in both years of observation. Considering the significant differences between the observation years in terms of climatic conditions (Figure 6) and observed scores (Figure 5), these genotypes could be valuable genetic material for breeding programmes. When we consider PH to highlight accessions that are less susceptible to lodging, we reduce this number to five accessions, namely ‘Sweden-1’, ‘Chishiminskaya’, ‘Dozhdik’, ‘Temnica na Krasu’ and ‘Nostrano’. Although these accessions have a high overall score across all the descriptors assessed, the scores for some descriptors may still be relatively low. For example, the TSW value for ‘Temnica na Krasu’ is below 20% of the maximum value in 2020, or the rutin value of ‘Sweden-1’ and ‘Chishiminskaya’ is below 20% of the maximum value in 2019.

## 3. Materials and Methods

### 3.1. Plant Material

136 common buckwheat accessions were used in this study (Appendix A). 120 accessions were provided by gene banks, nine were from commercial varieties, three were from the Czech Gene bank working collection and four were control varieties. In 2019 and 2020, all accessions were sown in two rows of 1 m length, 25 cm apart and 50 seeds per row. During the growing season, selected morphological and phenological traits were evaluated according to the List of Descriptor [30]. A representative sample of 10 g was taken from the harvest and delivered to the laboratory. The buckwheat seeds were stored in a dark and cool place (−18 °C) prior to further processing.

### 3.2. Weather Conditions

Figure 6 describes weather conditions in the locality of Prague Ruzyně in 2019 and 2020.

### 3.3. Chemicals

Standards of the phenolic compounds apigenin, caffeic acid, catechin, chlorogenic acid, epicatechin, gallic acid, hesperidin, hyperoside, isoorientin, isoquercetin, iso-vitexin, kaempferol, naringenin, orientin, procyanidines B1 + B3, and procyanidine B2, quercetin, quercitrin, rutin, vitexin, and the internal standard probenecid were purchased from Sigma–Aldrich (St. Louis, MO, USA). Methanol (LC-MS grade, ≥99.9%) was obtained from Riedel de Haën (Seelze, Germany). Formic acid (LC-MS grade, 99%) was purchased from VWR (Leuven, Belgium). Pure water was obtained from a Milli-Q purification system (Millipore, Bedford, MA, USA).

### 3.4. Standards Preparation and Sample Isolation

To prepare reference stock solutions, individual reference standards of phenolic compounds were dissolved in MeOH to obtain a stock solution of 0.5 mg/mL and stored at −18 °C. The stocks were further diluted with a methanol concentration range of 0.001–2.000 µg/mL to create calibration curves for phenolic compound quantification. Probenecid was dissolved in MeOH at 0.5 mg/mL to prepare a stock solution of the internal standard. Probenecid was then added into individual reference standard solutions or test samples to the final concentration of 0.1 µg/mL.

Buckwheat grains were pre-milled with an IKA A11 basic mill (IKA-werke, Staufen, Germany). They were then frozen in liquid nitrogen and ground using a mortar and pestle until a fine powder was obtained. The powder was placed in sealed plastic bags and stored in a cool place (−18 °C). The extraction procedure of phenolic compounds was based on a procedure modified from previous literature [63]. Briefly, samples of wholemeal flour from milled buckwheat seeds (0.1 g) were extracted twice with 1 mL of extraction solvent (80% methanol with probenecid as an internal standard at c = 0.1 µg/mL) in Eppendorf tubes for 60 min at 45 °C using an ultrasonic bath. Then the samples were centrifuged (15 min; 13,500 rpm; 25 °C) and corresponding supernatants were combined and filtered through 0.2−µm nylon syringe filters (Thermo Scientific, Rockwood, TN, USA). Extracts were stored at −18 °C prior to UHPLC-ESI-MS/MS analysis.

### 3.5. UHPLC-ESI-MS/MS Instrumentation

The chromatographic system (Dionex UltiMate 3000 UHPLC system, Dionex Softron GmbH, Germany) consisted of a binary pump (HPG-3400RS), an autosampler (WPS-3000RS), a degasser (SRD-3400), and a column oven (TCC-3000RS). Detection was performed on a quadrupole/orbital ion trap Q Exactive mass spectrometer (Thermo Fisher Scientific, San Jose, CA, USA). Analytes were separated on a reversed phase Ascentis Express C18 column (2.1 × 100 mm, 2.7 µm) from Supelco (Bellefonte, PA, USA). The LC-MS system was equipped with a heated electrospray ionization source (HESI-II) and Xcalibur software, version 4.0.

### 3.6. UHPLC-ESI-MS/MS Analysis

Chromatographic separation was performed using gradient elution with 0.2% formic acid in water as solvent A, and methanol with 0.2% formic acid as solvent B. Separation was started by running the system with 99% of solvent A + 1% of solvent B; followed by gradient elution to 40% A + 60% B at 11 min. The column was then eluted for 2 min at 100% of B. Equilibration before the next run was achieved by washing the column with 99% A + 1% B for 2 min. Total analysis time was 15 min. The column was maintained at 40°C at a flow rate of 0.35 mL/min and injection volume was 1 µL.

Ionization was run in the negative electrospray ionization (ESI) mode. Spray voltage was maintained at −2.5 kV. Sheath gas flow was 49 arbitrary units, auxiliary gas flow rate was kept at 12 arbitrary units and sweep gas flow was two arbitrary units. Capillary temperature was 260 °C. Nitrogen was used as sheath, auxiliary and sweep gas. Heater temperature was kept at 419 °C. S-lens RF level was 30. The mass spectrometer was generally operated in parallel reaction monitoring (PRM) mode. The precursor ions in the inclusion list were isolated within the retention time window ± 60 s, filtered in the quadrupole at isolation window (target *m*/*z* ± 0.8 *m*/*z*), and fragmented in an HCD collision cell. Product ions were collected in the C-trap at resolution 17,500 FWHM, an AGC target value of 1 × 10^6^ and maximum injection time of 50 ms. The normalized collision energy (NCE) was optimized for each compound. Precursor and daughter ions monitored, retention times and NCE values are shown in Appendix A. The accuracy and calibration of the Q Exactive Orbitrap LC-MS/MS was checked using a reference standard mixture obtained from Thermo Fisher Scientific. Data were evaluated by the Quan/Qual Browser Xcalibur software, v 4.0.

### 3.7. Determination of Phenolic Compound Concentration in Buckwheat Samples and Statistical Analysis

Identification of phenolic compounds in buckwheat samples (Appendix A) was based on their retention times relative to the authentic standards and on mass spectral data (accurate mass determination generating elemental composition and fragmentation patterns of a molecular ion) obtained by LC-MS, which were compared with those described in previous studies made on Orbitrap analysis of phenolic compounds [20,22,64].

Calibration curves were constructed by plotting the peak area (adjusted by probenecid as an internal standard) versus concentration of relevant reference standards. Data analysis was performed in Statistica 12 software (TIBCO software, Palo Alto, CA, USA).

### 3.8. Chemical Analyses

Kjeldahl analysis: Approximately 10 g of seeds from each tested accession was crushed in a grinding mill (IKA A11 basic, IKA^®^ Werke GMBH & Co.KG, Staufen im Breisgau, Germany) to create individual samples. The dry matter content of seed samples (5 g) was further dried in an electric hot-air drier at 105 °C for 4 h, according to the standard method CSN EN ISO 662 [65]. The content of crude protein from each sample was determined using the classic Kjeldahl mineralization method and calculated with conversion factor 6.25 [66].

Folin Assay: TPC was determined spectrophotometrically with Folin–Ciocalteau reagent. A modified method of Holasova et al. [67] was used. Two grams of lyophilized sample were extracted with 20 mL of 80% MeOH for 60 min in a centrifugation tube. The tubes were protected from sunlight by aluminium foil. The resulting extract (0.5 mL) was pipetted into a 50 mL volumetric flask and diluted with distilled water. Then, 2.5 mL Folin–Ciocalteau reagent (PENTA, Prague, Czech Republic) and 7.5 mL 20% sodium carbonate solution was added after agitation. After 2 h standing in the dark at laboratory temperature, absorbance at wave-length λ = 765 nm was measured against a blank on the spectrophotometer Thermo GENESYS™ 10UV UV-Vis (Thermo Scientific, Waltham, MA, USA). The results were quantified using gallic acid standard (Merck, Germany) and expressed as gallic acid equivalents (GAE).

DPPH Assay: The radical scavenging capacity (RSC) was determined on microtiter plates in MeOH extracts using the stable radical 2,2-diphenyl-1-picrylhydrazyl (DPPH) [68]. Briefly, 20 mL of MeOH was added to 1 g of sample and shaken for 90 min while it was protected from light by aluminium foil. Twenty µL of extract reacted for 10 min with 150 µL DPPH solution with initial absorbance A = 0.6 at 550 nm. The reaction occurred in the dark, and the absorbance at 550 nm was read afterward using a spectrophotometer (Sunrise absorbance reader, Tecan, Switzerland). The ability to scavenge the DPPH radical was determined using the standard curve obtained with Trolox (Sigma–Aldrich, Germany) in a range from 0.0 to 0.2 mmol/L. The results were expressed as Trolox equivalent (TE) AA.

The analyses were done in two repetitions for each sample.

### 3.9. Statistical Analyses

Each descriptor evaluated for a set of 136 buckwheat genotypes was measured in at least three biological replicates. Statistical analysis was performed mainly using the R program (R Development Core Team 2020) and Microsoft Office Excel v. 2016. Raw data for each descriptor were analyzed using Shapiro–Wilk normality test from the “stats” package. A two-way analysis of variance (ANOVA) was applied to the data to test whether there was a significant effect of year and genotypes on individual descriptor scores. To compare each accession with respect to each descriptor, the means along with the standard deviations for each descriptor were calculated separately for each accession and year of observation. Spearman’s rank correlation was also calculated for each pair of descriptors based on the mean values to test correlations between individual descriptors. To test whether the correlation coefficient was significantly different from zero, the correlation test function was applied. To evaluate individual accessions across all descriptors evaluated, a different approach was taken. For each descriptor except PH, the percentage of the maximum value of each descriptor was calculated separately for each accession and year of observation (Table 1). Based on these percentages, the score for each accession was calculated using the weighted average equation. Each percentage value was multiplied by a coefficient to magnify the value of the primary descriptors. The coefficient for the descriptors was set as follows:

Based on the above data, a heatmap was created using the ComplexHeatmap package, with percentages colour-coded on a scale from blue (0%) to red (100%). These were combined with data for score (weighted average) and plant height, both plotted on a categorical scale. Accessions were ranked in the heatmap according to the average score from both years of observation.

Data standardized by unit-variance scaling were also subjected to principal component analysis (PCA). The PCA data were then subjected to hierarchical clustering using a distance matrix based on Euclidean distances and the Average Linkage Clustering method. Routines within the NbClust package were used to determine the number of clusters for data from each year of observation. The results of the clustering were visualized using the ComplexHeatmap.

## 4. Conclusions

This study was conducted in the view of the current requirements of organic breeders and organic farmers to have new buckwheat varieties available, which are suitable for the environment of Central Europe. Since there is no available data on traits required/desired by breeders for organic/low-input agriculture, our activities of comprehensive phenotyping of buckwheat with potential for organic breeding will be the first source of information. For the first time, an extensive collection of 136 common buckwheat accessions was evaluated over two years under the conditions of the Czech Republic for selected morpho-agronomic characteristics, as well as for nutritional and medicinal composition. Using the UHPLC/HRMS/ MS method, 20 flavonoid metabolites were detected and quantified. It was confirmed that this method is rapid, sensitive and accurate and it has good repeatability. The content of these flavonoid metabolites varied between accessions and years. High levels of mainly flavonols (rutin and hyperoside), flavanols (epicatechin and catechin) and a group of flavones (vitexin, isovitexin, orientin and isoorientin) were confirmed in the accessions. The year 2020 proved to be more favourable for the formation of secondary metabolites with lower temperatures and higher rainfall and is related to the ability of plants to respond to different environmental conditions. On the other hand, CP content was similar between years and among accessions. Further, comparison and discussion of the obtained results with the previously published data showed the necessity and importance of evaluating broad buckwheat germplasm collections for different climatic conditions. Five accessions (‘Sweden-1’, ‘Chishiminskaya’, ‘Dozhdik’, ‘Notrano’, and ‘Temnica na Krasu’) exhibited excellent comprehensive performance and a high level of stability of evaluated traits under changing conditions during the two-year evaluation and can be considered as key components as parents or intermediate materials for breeding. From our results, it appears that the evaluated accessions of *Fagopyrum esculentum* are promising sources of valuable traits and nutrients.

## Figures and Tables

**Figure 1 plants-10-01262-f001:**
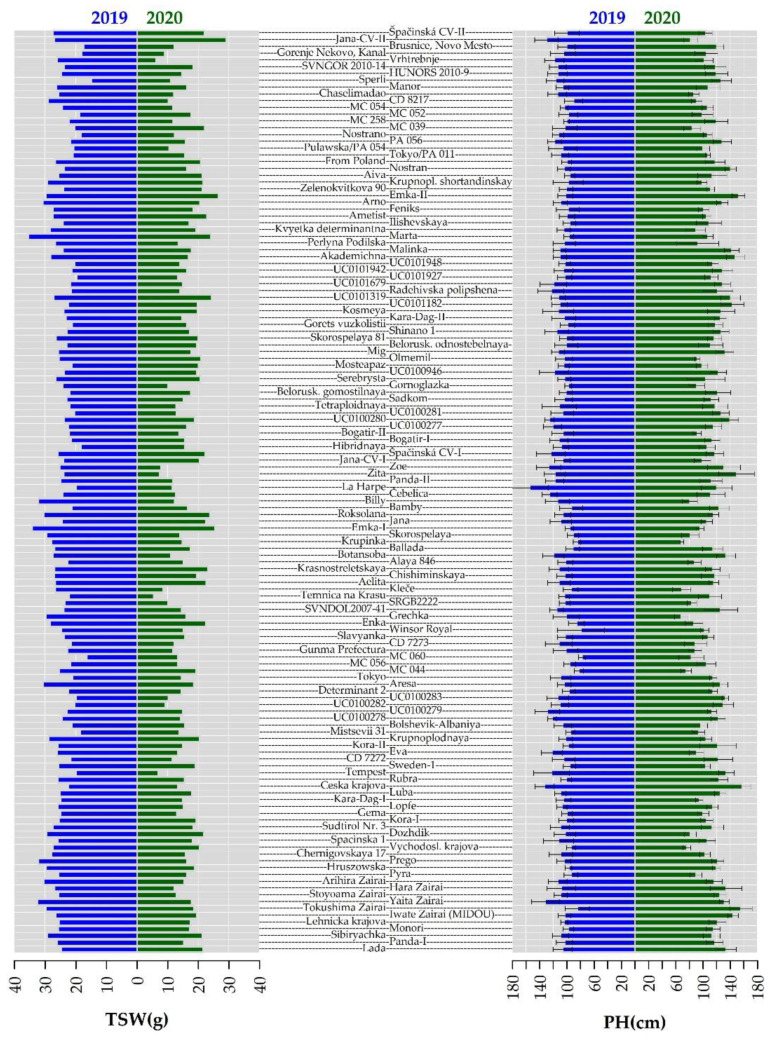
1000-seed weight (TSW) and plant height (PH) observed for collection of 136 buckwheat genotypes. Blue and green bars represent data from 2019 and 2020, respectively. The values are depicted as mean value from respective year ± standard deviation.

**Figure 2 plants-10-01262-f002:**
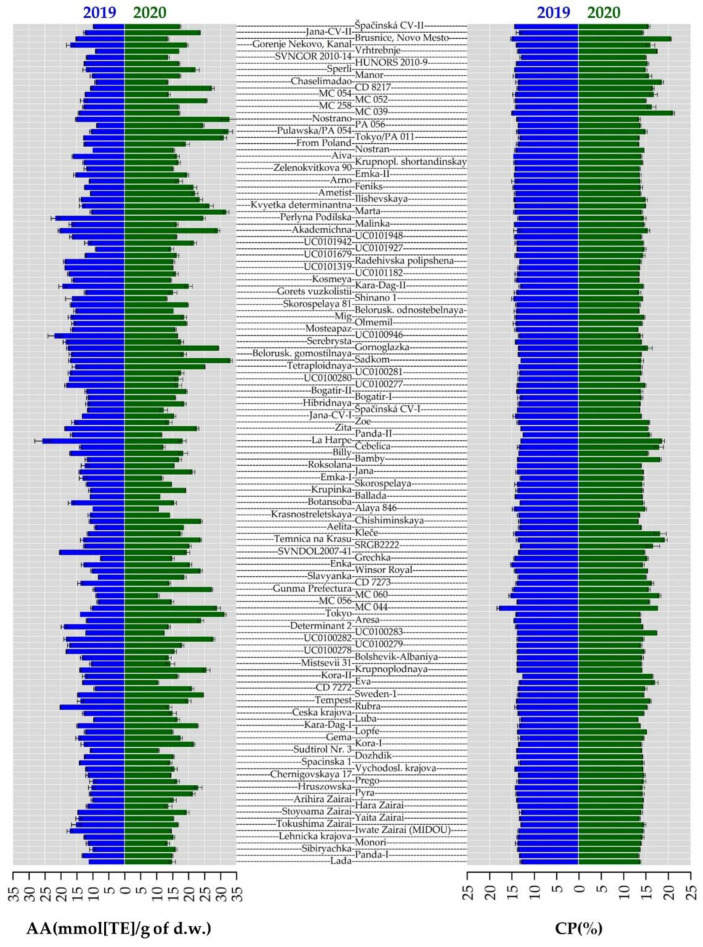
Antioxidant activity (AA) and crude protein content (CP) (AA) observed for the collection of 136 buckwheat genotypes. Blue and green bars represent the data from 2019 and 2020, respectively. The values are depicted as mean value from the respective year ± standard deviation.

**Figure 3 plants-10-01262-f003:**
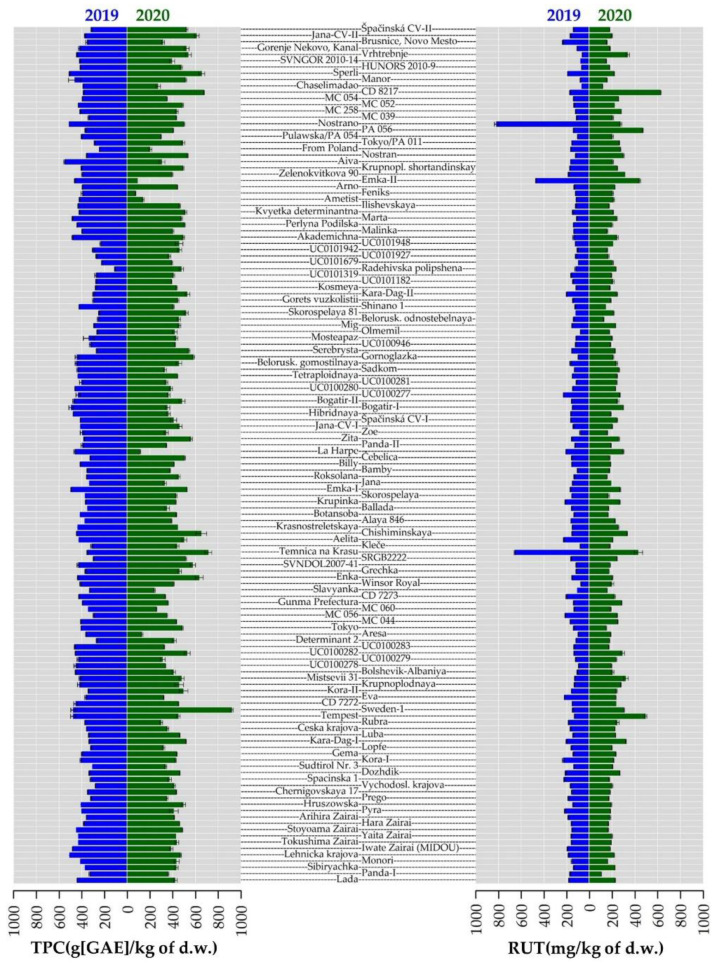
Total phenolic content (TPC) and rutin content (RUT) observed for the collection of 136 buckwheat genotypes. Blue and green bars represent the data from 2019 and 2020, respectively. The values are depicted as mean value from respective year ± standard deviation.

**Figure 4 plants-10-01262-f004:**
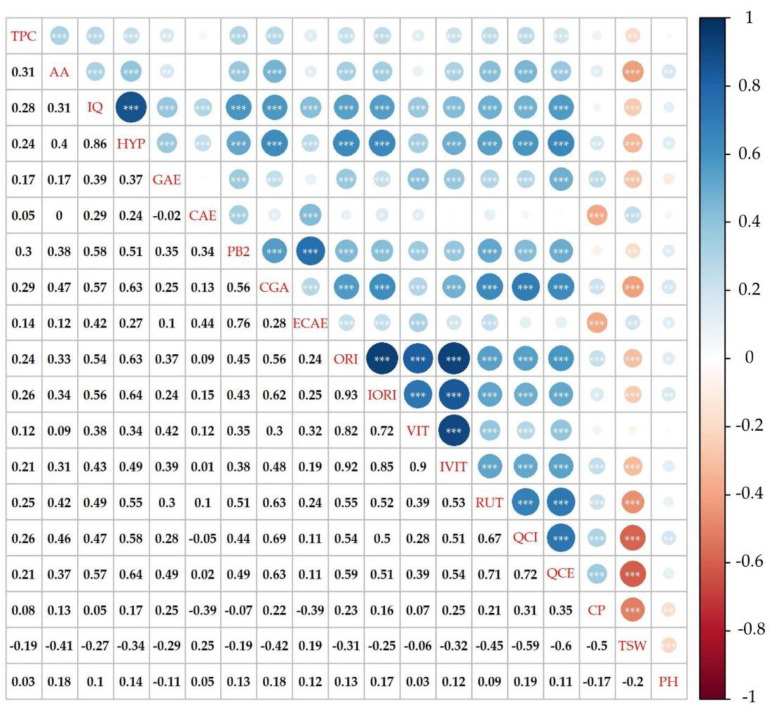
Spearman’s correlation between evaluated descriptors for the collection of selected buckwheat genotypes. The colours of the circles above the diagonal indicate whether the correlation between the pair of descriptors was negative (red)/positive (blue), while their magnitude is proportional to the Spearman’s ρ indicated below the diagonal. Significant correlations are represented by * (*p* < 0.05), ** (*p* < 0.01) and *** (*p* < 0.001), respectively.

**Figure 5 plants-10-01262-f005:**
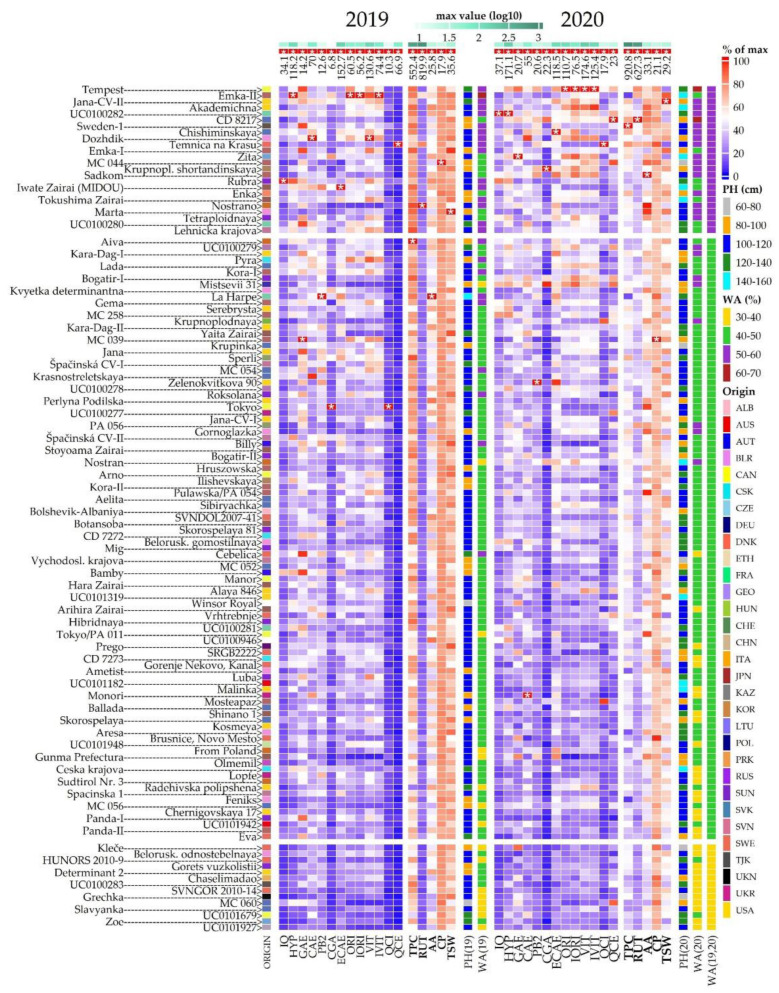
Heatmap showing levels of 19 descriptors (listed below) as observed for the set of 136 buckwheat accessions listed on left side of the plot. Mentioned descriptors comprise Plant height (PH [cm]), Antioxidant activity (AA [mmol TE/g d.w.), thousand seeds weight (TSW[g]), Crude protein (CP [% d.w.]) and content of total polyphenols (TPC [g GAE/kg d.w.]) and selected fractions, namely Isoquercetin (IQ), Hyperoside (HYP), Gallic acid (GAE), Cateching (CAE), Procyanidine B2 (PB2), Chlorogenic acid (CGA), Epicatechin (ECAE), Orientin (ORI), Isoorientin (IORI), Vitexin (VIT), Isovitexin (IVIT), Rutin (RUT), Quercitrin (QCI), Quercetin (QCE), all measured in µg/g d.w. Main heatmap bodies corresponds to values of 18 descriptors (excluding PH) depicted as percentage of maximal value (listed above the heatmap, marked with * ) recorded for respective descriptor in the given year of observation and displayed on a scale from blue (0) to red (100%). Side heatmap contains information about plant heights (PHs) for each accession as well as score based on weighted average (WA) of ratios to maximum values in each descriptor, displayed both on categorical scale. Accessions are sorted according to average score from both years (WA19, 20) and split into three groups with low, medium and upper score. Origin stands for country of origin of respective accession.

**Figure 6 plants-10-01262-f006:**
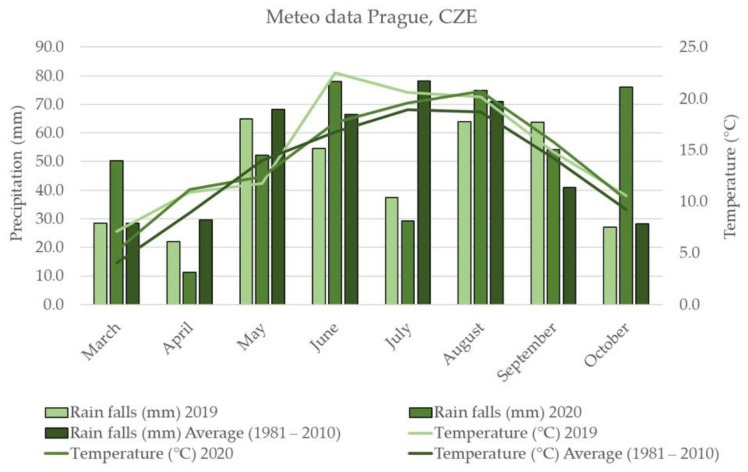
Weather conditions in the Czech Republic in 2019 and 2020 in comparison with the long-term average (1981–2010).

**Table 1 plants-10-01262-t001:** Coefficients for individual descriptors used in calculation of weighted average.

Type	Secondary Descriptors												Primary Descriptors					
Descriptor	IQ	HYP	GAE	CAE	PB2	CGA	ECAE	ORI	IORI	VIT	IVIT	QCI	QCE	TPC	RUT	AA	CP	TSW
Coefficient	1	1	1	1	1	1	1	1	1	1	1	1	2	4	3	4	4	4

GAE—Gallic acid, CAE—Catechin, HYP—Hyperoside, CGA—Chlorogenic acid, ECAE—Epicatechin, IORI—Isoorientin, ORI—Orientin, IQ—Isoquercetin, VIT—Vitexin, IVIT—Isovitexin, PB2—Procyanidine, RUT—Rutin, QCE—Quercetin, QCI—Quercitrin, TPC—Total polyphenols, TSW—1000-seed weight, AA—Antioxidant activity, CP—Crude protein.

## Data Availability

The data from experiments and analyses presented in this study are in a publicly accessible repository available in Appendix A.

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
