# Peer review of "Breeding Buckwheat for Nutritional Quality in the Czech Republic"

_plants, 2021, doi:10.3390/plants10071262_

Round 1
Reviewer 1 Report
The work “Breeding Buckwheat for Nutritional Quality in the Czech Republic” is very interesting from a scientific point of view. It is an accurate and interesting exploration of a large buckwheat germplasm collection.
The work is well organized and well written. A lot of data and an updated bibliography accompanies it. The results are well reported and strongly supporting the discussion. Results and Discussion are reinforce by numerous and appropriate bibliographic references.
However, I suggest the following changes:
- after the Introduction paragraph put Materials and Methods paragraph and subsequently Results and Discussions paragraph;
- I think that Weather conditions paragraph and Figure 6 may be part of the Results paragraph;
- Row 75 …… was one a third higher and of July where precipitation was half.
Author Response
Prague 9th of June 2021
Dear Ms. Li,
We are very grateful for your comments and valuable suggestions on our original manuscript entitled ´Breeding buckwheat for nutritional quality in the Czech Republic´ (No. plants-1249406). We would like to present our revised paper. We have considered all the reviewer’s comments (see below) and incorporated them into the manuscript. Changes in the text are highlighted within the manuscript text.
Sincerely yours,
Petra Hlásná Čepková on behalf of the authors
Responses to the reviewer’s comments and answers are listed below:
Answers to the all comments
Reviewer #3: The work “Breeding Buckwheat for Nutritional Quality in the Czech Republic” is very interesting from a scientific point of view. It is an accurate and interesting exploration of a large buckwheat germplasm collection.
The work is well organized and well written. A lot of data and an updated bibliography accompanies it. The results are well reported and strongly supporting the discussion. Results and Discussion are reinforce by numerous and appropriate bibliographic references.
However, I suggest the following changes:
- after the Introduction paragraph put Materials and Methods paragraph and subsequently Results and Discussions paragraph;
The structure of chapters is according to the publisher. Unfortunately, we cannot change it.
- I think that Weather conditions paragraph and Figure 6 may be part of the Results paragraph;
We agree with the reviewer.
- Row 75 …… was one a third higher and of July where precipitation was half.
The sentence has been modified according to the proposal of the reviewer.

Reviewer 2 Report
Breeding buckwheat for nutritional quality in the Czech Republic will be a valuable contricution to the journal plants. My concerns are as follows.
1. Restructure abstract and add more information.
2. Results are not comprehensively written and can be elaborated.
3. Discussion can be improved from examples for the literature and more references to relate the
results obtained.
4. Also update and replace old references with recent references
5. Check figures and their ligands.
6. Elaborate the discussion section with a correlation of the studies with your work in maybe other countries.
7. Restructure and carefully edit the conclusion section.
Author Response
Prague 9th of June 2021
Dear Ms. Li,
We are very grateful for your comments and valuable suggestions on our original manuscript entitled ´Breeding buckwheat for nutritional quality in the Czech Republic´ (No. plants-1249406). We would like to present our revised paper. We have considered all the reviewer’s comments (see below) and incorporated them into the manuscript. Changes in the text are highlighted within the manuscript text.
Sincerely yours,
Petra Hlásná Čepková on behalf of the authors
Responses to the reviewer’s comments and answers are listed below:
Answers to the all comments
Reviewer #1: Breeding buckwheat for nutritional quality in the Czech Republic will be a valuable contribution to the journal plants. My concerns are as follows.
- Restructure abstract and add more information.
Abstract has been changed and new information has been added with the aim of attracting more readers. Also, the main objective of our research as well as the main results has been included in the abstract with the plan of using the results.
Results are not comprehensively written and can be elaborated.
Results and discussion section was modified, more current and appropriate literature sources were used to discuss the results obtained in our study.
- Discussion can be improved from examples for the literature and more references to relate the results obtained.
Eight additional literature sources of current and recent articles have been added to the results and discussion section.
- Also update and replace old references with recent references
In the text, more than 30% of all cited references were not older than 2018, and as many as 50% were from the period between 2016 and 2021. However, more recent references (from 2021) were added where appropriate.
- Check figures and their ligands.
All figures and its ligands were checked and corrected.
- Elaborate the discussion section with a correlation of the studies with your work in maybe other countries.
The results and discussion section was elaborated and the obtained results were discussed with studies from other countries, but as we mentioned in the introduction there are only a limited number of studies whose results would be comparable to results of our study in terms of number of buckwheat accessions, number of phenols analyzed and the locality of Central Europe.
- Restructure and carefully edit the conclusion section.
The conclusion section has been reviewed and revised, and more summary and concluding information has been included in it.
Reviewer 3 Report
The aim of this study is to evaluate selected phenotypic and morphological traits using international buckwheat descriptors on 136 common buckwheat accessions grown in 2019-2020 and to determine protein, total phenolic content and antioxidant activity. The manuscript fits within the scope of the journal and is interesting. The title is clear and it is adequate to the content of the article. The revisions are necessary to improve the clarity of the presentation.
I have some questions/recommendations for authors:
Abstract:
Please highlight the degree of novelty and originality of the work.
Please include a general conclusion of your work.
Introduction
- Please include some information about the chemical content of buckwheat and biological activity.
Material and methods
DPPH Assay is original? If not, please include citations.
Results
Please include subchapter for Chemical Compounds analysis: CP, TPC, DPPH, UHPLC….
L523: Please use italic for the scientific name (Fagopyrum esculentum)
What are the future applications? What are the next research directions?
Author Response
Prague 9th of June 2021
Dear Ms. Li,
We are very grateful for your comments and valuable suggestions on our original manuscript entitled „Breeding buckwheat for nutritional quality in the Czech Republic“ (No. plants-1249406). We would like to present our revised paper. We have considered all reviewer’s comments (see below) and incorporated them into the manuscript. Changes in the text are highlighted within the manuscript text.
Sincerely Yours,
Petra Hlásná Čepková, Ph.D., corresponding author Dear Editor and Reviewers,
Responses to the reviewer’s comments and answers are listed below:
Answers to the all comments
Reviewer #2: The aim of this study is to evaluate selected phenotypic and morphological traits using international buckwheat descriptors on 136 common buckwheat accessions grown in 2019-2020 and to determine protein, total phenolic content and antioxidant activity. The manuscript fits within the scope of the journal and is interesting. The title is clear and it is adequate to the content of the article. The revisions are necessary to improve the clarity of the presentation.
I have some questions/recommendations for authors:
Abstract:
Please highlight the degree of novelty and originality of the work. – the novelty of the study was highlighted in the appropriate part of the manuscript (abstract, introduction, and conclusion parts)
Please include a general conclusion of your work. – the general conclusion was added to the abstract text.
Introduction:
Please include some information about the chemical content of buckwheat and biological activity. – the introductory part was substantially changed and the part about the chemical content of buckwheat and its biological activity was added to the text (lines 68-77). The other part of introduction (lines 95-103) has also been improved with up-to-date information.
Material and methods:
DPPH Assay is original? If not, please include citations. - Citation of DPPH Assay (Şensoy I. et al., 2006) was added to the text.
Results:
Please include subchapter for Chemical Compounds analysis: CP, TPC, DPPH, UHPLC…. The section of of the results and the discussion was divided into relevant chapters.
L523: Please use italic for the scientific name (Fagopyrum esculentum) – the scientific name was chenged to the correct form.
What are the future applications? What are the next research directions?
This study was conducted in the view of the current requirements of breeders and farmers to have available new buckwheat varieties suitable for the environment of Central Europe in the organic farming. Since there is no available data on traits required/desired by breeders for organic/low-input agriculture, our activities of comprehensive phenotyping of buckwheat with potential for organic breeding will be the first source of information. However, all results are usable also by other breeders. For the first time, an extensive collection of 136 common buckwheat accessions was evaluated over two years under the conditions of the Czech Republic for selected morpho-agronomic characteristics, as well as for nutritional and medicinal composition. Five accessions ('Sweden-1', 'Chishiminskaya', 'Dozhdik', 'Notrano', and 'Temnica na Krasu') exhibited excellent comprehensive performance and high level of stability of evaluated traits under changing conditions during the two year evaluation and can be considered as key components as parents or intermediate materials for breeding. From our results, it appears that the evaluated accessions of Fagopyrum esculentum are promising sources of valuable traits and nutrients.
Round 2
Reviewer 2 Report
The authors have incorporated the changes I mentioned; therefore, the manuscript can be accepted for publication after a careful spell check and correcting minor English language mistakes.